# Assessing the Expression of Long INterspersed Elements (LINEs) via Long-Read Sequencing in Diverse Human Tissues and Cell Lines

**DOI:** 10.3390/genes14101893

**Published:** 2023-09-29

**Authors:** Karleena Rybacki, Mingyi Xia, Mian Umair Ahsan, Jinchuan Xing, Kai Wang

**Affiliations:** 1Department of Bioengineering, University of Pennsylvania, Philadelphia, PA 19104, USA; krybacki@seas.upenn.edu (K.R.); mingyix@seas.upenn.edu (M.X.); 2Raymond G. Perelman Center for Cellular and Molecular Therapeutics, The Children’s Hospital of Philadelphia, Philadelphia, PA 19104, USA; ahsanm1@chop.edu; 3Department of Genetics, Rutgers, The State University of New Jersey, Piscataway, NJ 08854, USA; 4Human Genetics Institute of New Jersey, Rutgers, The State University of New Jersey, Piscataway, NJ 08854, USA

**Keywords:** transposable elements, TE, Long INterspersed Elements, LINE-1, long-read sequencing, gene expression

## Abstract

Transposable elements, such as Long INterspersed Elements (LINEs), are DNA sequences that can replicate within genomes. LINEs replicate using an RNA intermediate followed by reverse transcription and are typically a few kilobases in length. LINE activity creates genomic structural variants in human populations and leads to somatic alterations in cancer genomes. Long-read RNA sequencing technologies, including Oxford Nanopore and PacBio, can directly sequence relatively long transcripts, thus providing the opportunity to examine full-length LINE transcripts. This study focuses on the development of a new bioinformatics pipeline for the identification and quantification of active, full-length LINE transcripts in diverse human tissues and cell lines. In our pipeline, we utilized RepeatMasker to identify LINE-1 (L1) transcripts from long-read transcriptome data and incorporated several criteria, such as transcript start position, divergence, and length, to remove likely false positives. Comparisons between cancerous and normal cell lines, as well as human tissue samples, revealed elevated expression levels of young LINEs in cancer, particularly at intact L1 loci. By employing bioinformatics methodologies on long-read transcriptome data, this study demonstrates the landscape of L1 expression in tissues and cell lines.

## 1. Introduction

Transposable elements (TEs) are mobile DNA sequences that can move from one part of the genome to another and generate genomic diversity. Barbara McClintock’s discovery of TEs in the 1940s [1] marked the beginning of an extensive research journey aimed at understanding the mechanisms and functional effects of TEs [2,3,4]. In humans, TEs constitute at least 45% of the genome [5]. Their insertions significantly impact genomic stability [5,6,7,8,9], leading to a range of effects including developmental abnormalities [10], neurological disorders [11,12], and genetic diseases [3,13]. The mobilization of TEs can also result in chromosomal rearrangements and mutations at the insertion sites [14,15], further giving rise to genomic instability and cancer [16,17,18,19,20].

Among different types of TEs, the Long INterspersed Elements (LINEs) contribute to approximately 17% of the human genome and replicate using an RNA intermediate followed by reverse transcription, known as the “copy and paste” mechanism [21]. In the human genome, LINE-1 (L1) is the most active and abundant retrotransposon [5,6], with at least 500,000 copies of L1 elements present. However, only a small number of L1 elements maintain the coding capacity for the machinery required for retrotransposition [13,22]. Currently, L1s are the only autonomous retrotransposons that are still active in the human genome [23]. Autonomous L1 retrotransposition requires the expression and translation of its two open reading frames (ORFs), ORF1 and ORF2. ORF1 encodes a nucleic acid binding protein that binds to L1 RNA, while ORF2 encodes a protein with endonuclease and reverse transcriptase activities for retrotransposition [24]. At the transcription level, the polyadenylation signal is a critical component necessary for the retrotransposition as it facilitates the synthesis of a polyadenylated RNA intermediate, which is essential to the retrotransposition process [6,21,25]. Another important factor in L1 transcription is the upstream RNA promoter. L1s with an active promoter have the potential to be transcribed, which is necessary for L1 amplification [26].

LINEs have been present in the human genome for millions of years, accumulating mutations, rendering most inactive. While the majority of the older and mutated L1 elements have lost their ability to retrotranspose, younger and less divergent L1 elements with fully intact ORFs and polyadenylation signals retain the necessary machinery for retrotransposition and pose a greater risk for genetic instability [4,22,25,26,27]. The youngest L1 element with the ability to become reactivated for retrotransposition belongs to the L1HS (human specific) subfamily of L1 [27]. Previous studies have shown that L1 elements can be expressed and potentially active in somatic tissues, possibly contributing to somatic mutations and becoming potential cancer drivers throughout an individual’s lifetime [28]. Therefore, there is a need to study the contribution of TEs, particularly LINEs, to genomic instability and disease initiation/progression.

Conventional short-read sequencing techniques have severe limitations when studying TEs. In particular, TEs are highly repetitive in the genome and make it challenging to map the short reads to a particular TE when a short read matches to multiple TEs in the genome [29]. Although TEs are implicated in several diseases, their expression and activity at the tissue level remain understudied due to these limitations of short-read sequencing [30]. Long-read sequencing platforms such as Oxford Nanopore Technologies (ONT) and Pacific Biosciences (PacBio) are particularly advantageous for analyzing TEs in terms of sequencing read length and mappability. Compared to short-read sequencing, long-read sequencing can provide more detailed information on the full length of L1 elements, including the sequences of the 5′ and 3′ untranslated regions (UTRs), ORFs, and polyadenylation signals [26]. Furthermore, somatic TE insertions can be identified, which are often missed by short-read sequencing due to their low frequency and high complexity [31]. This potential for enhanced sensitivity and specificity point to long-read sequencing as an important tool for studying the activity and impact of L1 retrotransposons on genome stability.

In this study, we implemented a novel bioinformatics approach to assess the transcription of active L1 elements across multiple cell lines and somatic tissues. Our strategy involved utilizing RepeatMasker [32] to differentiate between the various repetitive elements in the genome, with a specific focus on the full-length and intact L1 elements in the L1Base2 database [33]. Using this approach, we investigated the expression levels of L1 elements in tissues and cell lines using different sequencing platforms, including ONT and PacBio, with different library preparation and sequencing protocols. Overall, this study contributes a novel bioinformatics pipeline for the analysis of TEs in long-read RNA-seq data, which will be helpful in improving our knowledge of the distribution and activity of TEs in the human genome.

## 2. Materials and Methods

### 2.1. Datasets

The number of samples and the sequencing platforms used in each dataset are detailed in Figure 1. The long-read RNA sequencing dataset from the Genotype-Tissue Expression (GTEx) project (v9) is composed of a total of 88 different tissue and K562 cell line samples, generated using the ONT sequencing platform (dbGAP accession number phs000424.v9) [34]. The main tissue types include adipose, brain, breast, cultured fibroblast cells, heart, liver, lung, muscle, and pancreas. Tissue subtypes include brain: anterior cingulate cortex, caudate, cerebellum, frontal cortex, and putamen; heart: atrial appendage and left ventricle; and cultured fibroblast: with and without protein PTBP1 knockdown. The K562 cell line originated from a patient with chronic myelogenous leukemia [34].

In addition to the GTEx tissue and cell line samples, several long-read RNA sequencing cell line datasets from both ONT and PacBio platforms were included in the study. The ONT sequenced datasets included the following cell lines: Universal Human Reference (UHR), HepG2, Human Embryonic Kidney 293T (HEK293T), HCT116, MCF7, A549, HeLa, and Acute Myeloid Leukemia (AML). The UHR dataset consisted of ten human cell lines derived from various human tissues, including liver, testis, mammary gland, cervix, brain, skin, liposarcoma, macrophage, T-lymphoblast, and B-lymphocyte (BioProject Accession Number: PRJNA639366) [35]. Datasets of five cell lines were obtained from the Singapore Nanopore Expression (SG-NEx) project [36] (accessed on 2 June 2023 at registry.opendata.aws/sg-nex-data): HepG2 cell line of human liver carcinoma, HEK293T cell line derived from human embryonic kidney cells that were transformed with adenovirus E1A and SV40 large T antigen [37], HCT116 cell line originated from human colorectal carcinoma, MCF7 breast cancer cell line from a patient tissue sample, and A549 lung cancer cell line from a patient tissue sample. The HeLa cell line originated from cervical adenocarcinoma (BioProject Accession Number: PRJNA777450) [38], and the AML cell line originated from a patient with AML, a form of blood cancer characterized by the production of abnormal myeloblasts in the bone marrow [39] (BioProject Accession Number: PRJNA640456) [36].

The PacBio sequenced datasets included the following cell lines: UHR (Accessed on 2 June 2023 and available online at https://downloads.pacbcloud.com/public/dataset/UHR_IsoSeq/) [40], HepG2 (PacBio Encode Accession Number: ENCFF483HTA), HCT116 (Gene Expression Omnibus: GSM5331110), K562 (Gene Expression Omnibus: GSE175347), and Esophageal Squamous Cell Carcinoma—ESCC (Gene Expression Omnibus: PRJNA515570). The ESCC samples used in this study included cancerous KYSE140, derived from a patient with moderately differentiated squamous cell carcinoma; KYSE510, derived from a patient with well-differentiated squamous cell carcinoma; TE5, derived from a patient with poorly differentiated squamous cell carcinoma (two sequencing runs); and an SHEEC cell line established from malignant transformation of the SHEE normal cell line induced by 12-O-tetradeeanoyl-phorbol-13-acetate (TPA) [41]. The normal ESCC cell line sample, SHEE, was an HPV18 E6E7-immortalized human embryonic esophageal epithelial cell line [41]. Appendix A provides a comprehensive overview of the sample references and preparation methods, along with their corresponding sequencing library construction protocols and sequencing platforms.

### 2.2. Data Preprocessing and Analysis Methods

An overview of the analysis pipeline employed in this study to characterize the expression of young L1 elements is presented in Figure 2. Detailed information and the code of the pipeline used in this study are available at https://github.com/WGLab/LINE-Expression-LRS. For raw datasets that were in the FAST5 format, FAST5 files were basecalled using Guppy6 [42] to generate FASTQ files. For raw datasets in FASTQ format, the FASTQ files were directly processed through the pipeline. For each dataset, sequencing reads in FASTQ files were filtered with the default base quality Q-score filtering threshold of 7. Replicates from the same biological sample were further concatenated. The FASTQ files were then converted into a FASTA file and sequence reads that were less than 1 kb in length were filtered out. This step helped to improve performance as these shorter reads are unlikely to be relevant to the analysis of LINE/L1 elements that typically range from 4 to 6 kb in length. The number of initial reads and reads after the size filtering are presented in Appendix A. To select reads that contain LINEs, reads were mapped to a custom library of LINE elements. The custom LINE library was generated by combining (1) LINE elements greater than 4.5 kb in length from the UCSC Genome Browser RepeatMasker Track [43] of the human reference genome (hg38); and (2) the LINE consensus sequences in the RepeatMasker database library. The sequence reads with lengths greater than 1 kb were mapped to this custom LINE reference using Minimap2 (v2.24) [44] (*minimap2 -ax map-ont “L1_reference.fasta” “input_reads.fasta” > “LINE_input_reads.fasta”*) to obtain reads that have at least one mapping to a LINE element. The resulting FASTA file was analyzed with RepeatMasker (version 4.1.5 released March 2023) [32] for LINE/L1 annotation. RepeatMasker was ran using the default library, specifying the species as human. The following feature checks were skipped: the bacterial insertion element, low complexity DNA or simple repeats, and small RNA (*RepeatMasker -pa 6 -dir “output_directory” -nolow -norna -species human -no_is -a -u -xsmall -xm “LINE_input_reads.fasta”*). Using the RepeatMasker output, sequences containing L1 elements that were identified as less than 10% diverged from the consensus sequence, as determined by the “div” column, were selected for the downstream analysis.

Reads containing L1 elements with less than 10% divergence were then mapped to the hg38 reference genome using Minimap2 with splice-aware parameters. A reference junction BED file (GENCODE.v40 [45]) was used to provide information about the known junction sites or splice sites within the reference genome. Only primary alignments were kept for downstream analyses (*minimap2 -ax splice --junc-bed reference_splice_junctions.bed -uf --secondary=no -k14 -t 8 hg38_reference.fasta input_reads.fasta -o aligned_reads.sam*). The generated SAM file was then converted to a BAM file using *samtools view*, after which the BAM file was sorted according to genomic location and indexed. *LongReadSum* (v1.3.0) [46] was used for the quality check analysis of the mapping result (*python LongReadSum bam -i sample_input.bam*) and the following statistics are presented in Appendix A: initial total number of reads, N50 read length, median read length, number of reads greater than 1 kilobase (kb), total number of reads following RepeatMasker, total number of mapped reads (primary alignments), N50 mapped read length, median mapped read length, and the total number of reads following artifact-based filtering.

The L1Base2 (v2) database for the human genome (*Homo sapiens*) was generated from the GRCh38 assembly, where the consensus sequence for different subfamilies and details about the existence of ORFs are reported [33]. L1Base2 consists of three BED files with genomic regions of three different L1 categories: (i) full-length L1s with intact ORF1 and ORF2 (FLI-L1s) (146 regions, referred to as “Active” in Figure 2), (ii) full length non-intact L1s (FLnI-L1s) (13,418 regions, referred to as “Inactive” in Figure 2), and (iii) L1s with a disrupted ORF1 and an intact ORF2 (ORF2-L1s) (107 regions, referred to as “ORF2” in Figure 2) [33]. An initial filtering process was performed on these files using *bedtools intersect* [47] to generate three distinct reference files for the different categories of L1 elements. For the active elements (FLI-L1s), any regions with overlapping occurrences in the inactive (FLnI-L1s) elements were removed. Additional filtering was completed for downstream artifact-based filtering, where L1 regions overlapping exons were excluded (GENCODE.v43 [45], *bedtools intersect -a “reference_regions.bed” -b “GENCODE.v43.annotation_exons.bed” -v > “reference_no_exons.bed”*). As a result, three distinct reference files were obtained: active L1 elements (FLI-L1s) (115 regions), inactive L1 elements (FLnI-L1s) (10,993 regions), and L1 elements intact only in ORF2 (ORF2-L1s) (97 regions). The purpose of this filtering was to ensure that each reference file contained specific and non-overlapping regions for subsequent L1 analysis. Each resulting reference file (FLI-L1, FLnl-L1s, and ORF2-L1s) was used to determine the coverage of the different types of L1 elements within the samples. Reads where less than 90% of the read was mapped to the L1 regions were removed (*samtools view -b -L “reference_regions.bed” -o “filtered_reads.bam” “mapped_reads.bam”, bedtools intersect -a “filtered_reads.bam” -b “reference_regions.bed” -f 0.9 > “filtered_reads_90percent.bam”*). By retaining reads with at least 90% mapped to the L1 loci, we ensured the retention of reads primarily aligned within the L1 regions of interest. Once the number of mapped reads for each locus was determined, reference L1 loci with fewer than two mapped reads were also excluded. For reads mapped to the remaining L1 loci, the start positions of the reads within each L1 region were compared. L1 loci with variable read starting positions that were more than 100 bps from the most common read start position were removed. Finally, the overall most common start positions of the reads within an L1 region were compared with the L1 region’s start position. An L1 loci was excluded if the most common read start position was more than 1.5 kb away from the reference L1 start position. This filtering approach further improved the accuracy and reliability of L1 analysis by filtering out potential false positives from possible technical biases due to library preparation, polymerase chain reaction (PCR) amplification biases, sequencing errors, and mapping errors.

Following the read filter portion of the artifact-based filtering step, the filtered BAM files were processed through the *bedtools genome coverage* function [47] (*bedtools genomecov -ibam “input_reads.bam” -bga -split > “coverage.bg”*). This resulted in a bedgraph, which has varying read count intervals of the coverage values across the L1 loci. Then, the *bedtools map* function [47] was used to determine the coverage of the L1 loci (*bedtools map -a reference_L1.bed -b coverage. bg -c 4 -o mean -null 0 > mean_coverage.txt*). Here, the *bedtools map* function calculated the mean number of RNA sequencing reads that intersected with the coordinates of the L1 loci from the L1Base2 reference. To determine the general expression level of these L1 loci, the calculated coverage values were normalized by the total number of reads in the sample to account for variations from the multiple datasets with varying total number of reads. A weighted average L1 expression value was then computed across all L1 loci in each of the reference L1Base2 categories: active, inactive, and intact only in ORF2, for each sample. This weighted average involved assigning weights based on the length of the reference L1 element with its corresponding expression value, scaled in millions, as displayed in Equation (1), where *i* takes the L1 reference category, “active”, “inactive”, and “intact only in ORF2” to represent the different categories.
(1)$$Weighted\;Average_{i}=\frac{\sum_{Regions_{i}}\left(Mean\;Number\;of\;RNA\;Sequencing\;Reads\;\cdot Length\;of\;L1\;Element\right)}{\sum_{Regions_{i}}\left(Length\;of\;L1\;Element\right)}\;For\;i\in \left\{Active,\;Inactive,\;Intact\;only\;in\;ORF2\right\}$$

Subfamily names of L1 elements longer than 4.5 kb in length from the UCSC Genome Browser (hg38) RepeatMasker Track, namely L1HS and L1PA2, were used to determine the L1 subfamily breakdown in the active, full-length L1 loci. Manual review using the Integrative Genomic Viewer (IGV) [48] was performed to visually validate the L1 loci with expression. This assessment aimed to identify any additional technical artifacts that might not have been accounted for in the bioinformatics pipeline originally.

## 3. Results and Discussion

The current study developed a bioinformatics pipeline leveraging long-read sequencing technologies to investigate the transcription patterns of L1s in human tissue and cell line samples. The degree to which L1 elements contribute to human genome function varies depending on several factors, including the tissue-specific regulation, the expression level of L1s, and the specific genomic locations of their transcription and subsequent insertions. Building upon previous work in L1 biology, we used long-read RNA sequencing datasets from ONT and PacBio to examine tissue-specific expression patterns of L1 elements.

### 3.1. General Description of the Datasets

Among the datasets used in this study (Figure 1), 88 tissue and cell line samples from the GTEx dataset were sequenced by ONT. Additionally, we included five cell lines sequenced exclusively with ONT technology, six cell lines sequenced exclusively with PacBio technology, and four cell lines sequenced using both ONT and PacBio technologies. Our dataset included samples subjected to a diverse array of RNA extraction and sequencing methods, as described in Appendix A. This diversity includes incorporation of the poly-A selection during the RNA sample preparation, which can have an influence on the RNA captured and downstream analysis. Samples prepared with poly-A selection included A549, HCT116, HepG2, and MCF7 sequenced by ONT, and HCT116, HepG2, and K562 sequenced by PacBio. Furthermore, the variety within the dataset extended to the library construction protocols, where some samples underwent direct RNA sequencing (ONT sequenced: A549, HCT116, HEK293T, HeLa, HepG2, MCF7, and UHR), while others followed a cDNA-based approach (ONT sequenced: GTEx and AML, PacBio sequenced: ESCC and UHR). The initial number of reads among the samples varied from 46,331 (Nanopore sample GTEX-T5JC-0011-R10A-SM-2TT23.FAK91589) to 13,363,390 (Nanopore sample GTEX-WY7C-0008-SM-3NZB5_ctrl.FAK55628), as detailed in Appendix A. The dynamic landscape of the read length is reflected in the N50 and median read length ranges, which extended from 585 to 3148, and 198 to 2623, respectively. Both the lower bounds of the N50 and median read length ranges originated from the ONT sample GTEX-UTHO-2426-SM-38ZXF.FAK46748, while the upper bound was from the PacBio sample ESCC SRR8691125. After filtering out reads that were less than 1 kb long, the number of reads in each library varied from 9760 (ONT sample GTEX-T5JC-0011-R10A-SM-2TT23.FAK91589) to 5,839,327 (PacBio sample ESCC SRR8490237). In our study we recognized the variety of differences in the samples’ origin, as well as the sample preparation and sequencing techniques. Throughout our analysis, we anticipated that these variations would inherently result in variable L1 expression levels, since these technical variations can have an influence on recapitulation of L1 transcripts that are longer than typical mRNA transcripts. We acknowledge these limitations due to technical heterogeneity of the various datasets; however, we utilized them to help provide a comprehensive overview of each sample’s genomic landscapes. With the evolving landscape of long-read sequencing (both in library construction protocols and in sequencing platforms), we also anticipate that future sequencing studies may include longer transcripts (such as L1) in the sequencing data that were not effectively captured in the current study due to the limitations of the technologies. With these caveats in mind, these technical variations contribute to the uniqueness of our datasets and helped facilitate a diverse array of insights into L1 detection and analysis.

### 3.2. Overall Expression Levels of L1s

An overview of the analysis pipeline is presented in Figure 2. The initial sequencing data were preprocessed, filtered, and mapped to a custom LINE reference to select candidate reads that were longer than 1 kb and contained L1 sequences. We then used RepeatMasker to identify sequencing reads containing L1s with less than 10% divergence and mapped these reads to the hg38 reference genome (mapping statistics in Appendix A). We then calculated the coverage, scaled in millions, within the active (FLI-L1s), inactive (FLnl-L1s), and intact only in ORF2 L1s (ORF2-L1s) in each tissue and cell line sample. A total of 88 human samples from eight major tissue types and two types of cell line (fibroblasts with or without PTBP1 knockdown) were examined by the ONT sequencing platform. Figure 3A provides an overview of the overall non-zero coverage across different human samples from the GTEx project [34]. The tissue types with zero coverage, breast, pancreas, and subcutaneous adipose, were removed from further analysis. The brain and liver tissues exhibited the highest expression levels from active L1s, while all other tissues did not show detectable expression levels of active L1s. Several samples showed relatively low coverage for inactive L1s. Additionally, only the brain tissues displayed expression of intact only in ORF2 L1s.

We further examined the distribution of the human tissue subtypes in brain and heart (Figure 3B) and observed a higher abundance of coverage within L1 regions for the different brain tissue subtypes, in comparison to the different heart tissue subtypes. Cerebellar Hemisphere and Putamen (Basial Ganglia) had higher coverage within the active L1 regions, while the Anterior Cingulate Cortex, Frontal Cortex, and Caudate had no expression. The heart tissue subtypes showed no expression in the active L1s and intact only in ORF2, and minimal expression within the inactive L1 regions. The expression level of the different categories of L1 elements in the tissue subtype samples remained relatively consistent, showing no apparent difference in L1 abundance among tissue subtypes. The low coverage values of active, intact only in ORF2, and inactive L1s suggest an overall low level of L1 transcription in human somatic tissues.

There are several caveats to the analysis: first, because existing long-read RNA-seq library and sequencing protocols are geared towards typical transcripts (~1.5 kb length on average), they may not be optimal for the analysis of L1 elements (~6 kb length) as only part of the L1 transcript sequences may be present in the data. Second, we used a relatively stringent bioinformatics protocol and removed a portion of the candidate L1 transcripts from the final analysis to focus on the most reliable subset of reads arising from L1 elements in the analysis. Third, the “less than 10% diverged” was a relatively arbitrary threshold to focus our analysis to recently diverged elements, but the ~3% sequencing error rate of ONT (when sequenced with R9 flowcell and basecalled by Guppy6 [42]) and the repetitive nature of the L1 elements may have influenced the effectiveness of this threshold. With the recent introduction of R10.4.1 flowcell with sequencing kit V14, the sequencing error rate has dropped to 1%. Lastly, the large variation in the sequencing depth among GTEx samples might have affected the sensitivity of L1 expression detection sensitivity among samples.

Next, we extended our analysis to include a variety of cell lines. Cell lines sequenced by PacBio are shown in Figure 4A, including UHR, HepG2, K562, and five ESCC lines. All cell lines except HCT116 showed expression of active L1s at varying levels. Three ESCC cell lines (KYSE510, KYSE140, and TE5) showed the highest expression among the cell lines. Cell lines sequenced by ONT included HCT116, HepG2, UHR, K562, HEK293T, HeLa, AML, MCF7, and A549. For ONT datasets, only AML showed L1 expression across the inactive regions (6.0) and intact only in ORF2 regions (1.0). Among the four cell lines (HCT116, HepG2, UHR, and K562) that were sequenced by both ONT and PacBio, HepG2, UHR, and K562 showed L1 expression as sequenced by PacBio but not by ONT (Appendix A), likely due to the much higher sequencing coverage in the PacBio sequencing data (Appendix A). Furthermore, focusing on specific L1 subfamilies that are present in the active L1 category, L1HS and L1PA2, Figure 4B displays a heatmap of the average number of expressed active L1 regions for these subfamilies in cell line samples. L1HS and L1PA2 are among the most active L1 elements [49] and had shown evidence of expression. The number of active L1 loci with expression from both L1HS and L1PA2 subfamilies was generally lower in the ESCC SHEE (Normal), HepG2, and K562 cell lines, compared to UHR and the ESCC cancer cell lines (Figure 4B). In particular, the cancerous ESCC cell lines displayed the highest number of active L1 loci with expression, exceeding that of the normal ESCC cell line. The identification of L1HS expression in the cell lines further supports the need for targeted investigations into the functional impact of L1 elements in various cellular contexts.

### 3.3. Expression Levels within L1 Regions with High Coverage and Manual Evaluation in IGV

To further confirm that our results reflect authentic L1 transcription, we manually evaluated L1 expression levels and patterns at the identified L1 regions using the Integrative Genomics Viewer (IGV) [48]. We examined various aspects, such as read alignment pattern, depth, and coverage distribution, to gain insights into the transcription dynamics among the L1 elements with the evidence of expression. In the manual evaluation process, we closely examined the coverage distribution across over 100 different high coverage L1 regions across the three L1Base2 reference categories—active, inactive, and intact only in ORF2—in different samples. This enabled the identification of areas with consistently high coverage that may signify increased expression levels. Similar to previous studies, we adopted the assumption that the transcripts started at or near the 5′ end of the L1 locus are likely to be from autonomous transcription of the L1 element [50]. We integrated this criterion into the manual review specifically for samples exhibiting high coverage to identify robust transcriptional activity. In IGV, we also included tracks of the hg38 reference genome, GENCODE.v43 annotation [45], and L1Base2 active, inactive, or intact only in ORF2 L1 reference regions [33].

Various alignment patterns were observed, including almost full coverage, reads concentrated at the 5′ or 3′ ends, and reads distributed throughout the middle of the region. Figure 5 displays an active L1 region, chr22:28,662,282-28,670,329 (hg38), which showed expression among the four cancerous ESCC cell line samples. By inspecting the alignment patterns across the different ESCC samples, we found that the samples had regions of coverage towards the 5′ end of the L1 (111 bps downstream of the L1 region start position) and KYSE510 showed almost complete coverage of the L1 locus. Among the ESCC samples, SHEEC and TE5 showed higher mismatch patterns than KYSE140 and KYSE510. This difference is likely attributed to the characteristics of PacBio IsoSeq data. Notably, the first two samples displayed fewer mismatches due to the shorter fragments in their initial 1 kb regions. Shorter reads possess higher accuracy from the IsoSeq sequencing, benefiting from the consensus-based error correction during the IsoSeq analysis. Alternatively, the mismatch patterns in the bottom three samples may suggest the transcripts mapped to this locus originated from polymorphic L1 elements that are not represented in the reference genome. Furthermore, Appendix A shows the alignment patterns of reads within the ESCC KYSE140 cancer cell line at an inactive L1 region (Appendix A, chr6:57,284,305-57,294,481, hg38) and an intact only in ORF2 L1 region (Appendix A, chr10:85,354,505-85,362,552, hg38). Analyzing these alignment patterns was crucial in confirming the presence, structure, and precise location of the L1 transcripts. The evidence of L1 expression from multiple active L1 loci in the ESCC samples corroborates with previous studies that highlighted the possibility of L1 elements being involved in esophageal squamous cell carcinoma through the examination of aberrant methylation patterns [51,52]. These results support the functional relevance of L1 components in ESCC and call for future research into how L1 transcription and somatic retrotransposition can influence cancer progression.

In addition to the ESCC samples, the UHR dataset sequenced using PacBio exhibited high coverage values in a few active L1 regions. Over the identified regions, the supporting reads revealed a consistent pattern of aligning towards the 5′ end, as displayed in Appendix A for two active L1 loci (1003 and 877 bps downstream of the L1 region start position, respectively). This suggested the presence of transcriptional initiation sites within these regions, where the transcription machinery initiates L1 transcription. The distribution of reads consisted of those extending only to the 5′ UTR of the regions and some reads mapped to the middle of the L1 regions. This heterogeneous distribution of reads indicated potential alternative transcriptional initiation and termination sites, including internal transcriptional activity within the L1 regions. Alternatively, this observation may represent sequencing artifacts of incomplete transcripts due to the relatively large transcript sizes. Overall, we manually evaluated 75 active L1 expression loci in different samples, where all 40 unique loci showed patterns of authentic L1 transcription (Figure 4B, Appendix A). This result suggested that by employing the rigorous artifact-based filtering, we had eliminated most of sequenced reads that did not derive from L1 transcription. This resulted in high confidence in identifying regions with authentic L1 expression for reliable L1 expression analysis.

Our analysis also examined the L1 expression patterns in human tissues, where the brain samples had the highest level of L1 expression. Analyzing the IGV plots from the brain sample provided evidence that suggested the expression of active L1 elements (Appendix A). Appendix A is an active L1 region within three brain cerebellar hemisphere tissue samples. We also extended our analysis to inactive L1 loci, with an example shown in Appendix A. Our analysis of human tissue samples from the GTEx project revealed evidence of somatic L1 expression within the brain, including active L1s. Furthermore, the increased levels of L1 expression in human brain tissues are consistent with previous studies that showed L1 expression in diverse brain areas [53,54,55]. These consistent results add to the growing evidence that L1 retrotransposons potentially contribute to normal brain function and pathological conditions. While L1 retrotransposons are likely to play a role in the normal function of the human brain, it is important to note that their contribution to pathological conditions could be more substantial [56,57,58]. Our study did not examine brain samples from patients with neurological diseases, so we hope to perform a comparative analysis to study this aspect in the future.

In our analysis we noticed a wide variety of alignment patterns within the L1 regions, including consistently uniform expression of L1 loci at the 5′ end. The partial L1 coverage alignment pattern suggested that full-length coverage was limited to certain portions of the L1 regions, or that only a partial portion of full-length transcripts were sequenced due to technical limitations such as RNA degradation or lower efficiency of reverse transcription for longer RNAs during the sequencing library construction. The results further emphasized that the lack of concordance in profiling L1 elements remained, even after implementing a rigorous filtering strategy to exclude low abundance reads that could have potentially introduced artifacts or were prone to mapping errors. The assumption that the expression at the 5′ end of L1 transcripts can be considered as an estimate for autonomous transcription was influenced by previous studies and by the observed imbalance between 5′ truncated copies and those fuller-length copies, which aligns with findings from prior research [50]. Given additional confirmation of internal transcriptional activity, we also found reads distributed across the middle of the predefined regions. These alignment patterns shed light on the structure and position of the expressed L1 elements in the samples.

### 3.4. Diverse Sequencing Statistics and Their Differential Impact on L1 Profiling

One important aspect of our analysis was the utilization of data from both PacBio and ONT long-read sequencing technologies. Given the highly repetitive nature of L1 elements, the sequence similarity of different subfamilies, and the accumulation of mutations through time, longer reads from both technologies helped to assign the reads more accurately to their proper positions during genome mapping. However, it is important to note that ONT sequencing is known for its higher levels of basecalling error (compared to the PacBio HiFi protocol), which can impact mapping accuracy [59]. Although recent chemistry and R10 flowcell versions of ONT sequencing greatly improves basecalling accuracy, the data that we used in the current study were generated using the R9.4 flowcell with a higher level of basecalling errors.

Using the data from both ONT and PacBio platforms also allowed us to examine the sequencing statistics (Appendix A) and assess the impact of sequencing technologies on our analysis. These statistics highlight the differences in initial number of reads, number of reads remaining after data processing prior to mapping to the hg38 reference genome, read lengths, the number of primary and supplementary mappings, and number of reads remaining after artifact-based filtering. It is important to consider the impact of library construction and sequencing technique on the detection and quantification of L1 expression. The library construction varied between cDNA and direct RNA sequencing, which resulted in a varying number of sequenced reads overall. For example, ONT direct RNA sequenced HepG2 resulted in 1,537,237 reads while ONT cDNA sequenced AML resulted in 8,061,683 reads. The inherent differences in the library construction resulted in fewer reads for direct RNA sequenced HepG2; however, the median read lengths were 1696 and 613 bps for HepG2 and AML, respectively, as shown in Appendix A. This indicated that while cDNA was able to generate additional sequences, direct RNA sequencing was able to generate longer, full-length transcripts. Additionally, the initial number of reads within the UHR datasets sequenced by ONT and PacBio were vastly different from each other, i.e., 476,000 compared to 6,775,127 reads, respectively. Following the first filtering of reads that were less than 1 kb long, the remaining number of reads in the UHR datasets sequenced by ONT and PacBio changed to 211,577 and 5,374,170, respectively. As additional artifact-based filtering methods were applied, only the UHR dataset sequenced by PacBio had a subset of 37 reads remaining for L1 detection and analysis. To mitigate differences such as these and gain further insights into the expression patterns of younger L1s in the different samples, we analyzed datasets from both PacBio and ONT datasets. However, due to the lack of a correlation between the two sequencing platforms, the comparison between these regions was not applicable.

## 4. Conclusions

Our study provides insights into the expression patterns and characteristics of L1 elements in human tissue and cell line samples, specifically focusing on L1s that diverged less than 10%. We observed low expression levels of full-length L1s in normal tissues, except for the brain, while variable expression was found in cell lines, with higher levels observed in tumor cell lines. As one of our future research directions, we plan to examine L1 elements in putamen samples from Parkinson’s disease patients and compare these with L1 expression in normal putamen samples. This is motivated by our observation of substantial coverage of active L1 regions in the putamen, which is implicated in the pathology of Parkinson’s disease. The prominent expression of the active L1HS subfamily supports its potential role in transpositional mutagenesis in the context of disease [60,61,62]. The findings from this study reinforce the significance of studying L1 expression dynamics and their functional impact in normal brain tissues and cell lines. While datasets used in this study are generated at varying timepoints and technologies, it is important to note that there have been continuous advancements in long-read sequencing protocols and methodologies, and the data quality is improving constantly. Our L1 detection and quantification pipeline provided a means to evaluate L1 expression levels and patterns in full-length RNA-seq data. Through a set of stringent filtering criteria, the bioinformatics pipeline presented in the study can adapt to diverse types of long-read RNA-seq datasets generated by different library preparation protocols on different sequencing platforms. This is because the underlying principles and approaches for identifying and quantifying the expression of L1 elements remain relevant and applicable, regardless of the specific sequencing platform or improvements in protocols. Overall, our study contributes to the understanding of L1 expression patterns and highlights the importance of utilizing long-read platforms followed by a robust bioinformatics analysis pipeline with rigorous filtering methods to accurately assess L1 expression. Building upon the pipeline presented in this paper, researchers can incorporate these advancements and refine their analyses, taking advantage of the continuous improvements in sequencing techniques for the comprehensive analysis of transposable elements.

## Figures and Tables

**Figure 1 genes-14-01893-f001:**
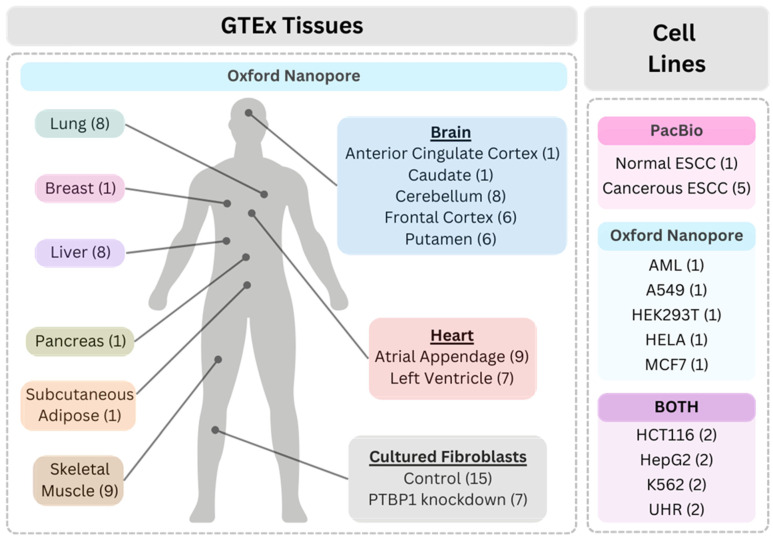
Overview of the datasets composed of human tissues, as well as normal and cancer cell lines. Tissues from the GTEx dataset (**left**) and cell lines (**right**) with corresponding number of samples. The GTEx dataset included 88 samples of tissues and fibroblast cell lines sequenced by Oxford Nanopore. Normal and cancer cell lines with corresponding number of samples are shown in the right panel. Cell lines sequenced with PacBio (pink), Oxford Nanopore (blue), or both platforms (purple) are shown in separate sections.

**Figure 2 genes-14-01893-f002:**
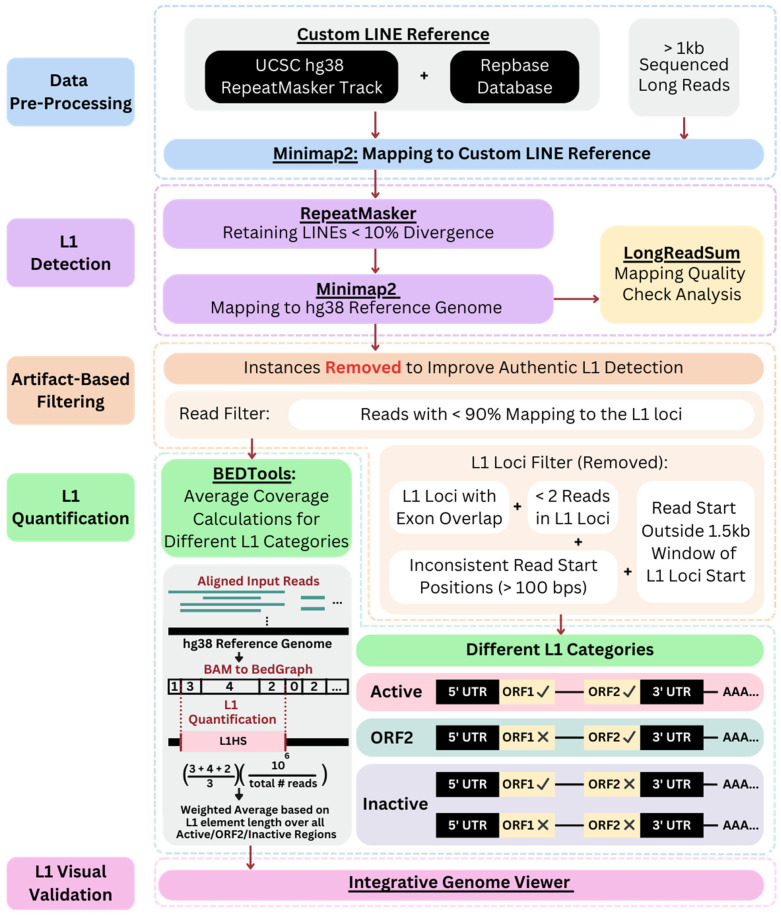
Overview of L1 analysis methods. Data Preprocessing: Reads less than 1 kb in length were removed. A custom LINE reference library was constructed by merging the UCSC hg38 RepeatMasker track with the LINE/L1 consensus sequences from the Repbase RepeatMasker library. The LINE custom library was used as a reference for mapping with Minimap2 to obtain input reads (>1 kb) containing L1 elements. L1 Detection: Reads with L1 elements with less than 10% divergence from the RepeatMasker L1 consensus were retained for Minimap2 splice-aware mapping to the hg38 reference genome. LongReadSum was utilized to report mapping statistics for quality check analysis. Artifact-Based Filtering: First, we removed reads where less than 90% of the read itself mapped to the L1 region of interest. Then, L1 regions with exon overlap, fewer than two mapped reads, inconsistent read start position among the reads within the L1 region (threshold 100 bps), and overall starting positions too far from the L1 region start position (threshold 1.5 kb) were removed from analysis to filter out likely false positives. L1 Quantification: BEDTools genome coverage and BEDTools map were used to calculate the average coverage over the three groups of L1 elements, including active, inactive, and those intact only in ORF2. The resulting values were first normalized by the total number of reads and then scaled by a million. Subsequently, a weighted average was computed, considering the length of the L1 element, across all active, intact only in ORF2, and inactive L1 loci, respectively, resulting in the final value. L1 Visual Validation: L1 candidate loci were visually validated in the Integrative Genomic Viewer.

**Figure 3 genes-14-01893-f003:**
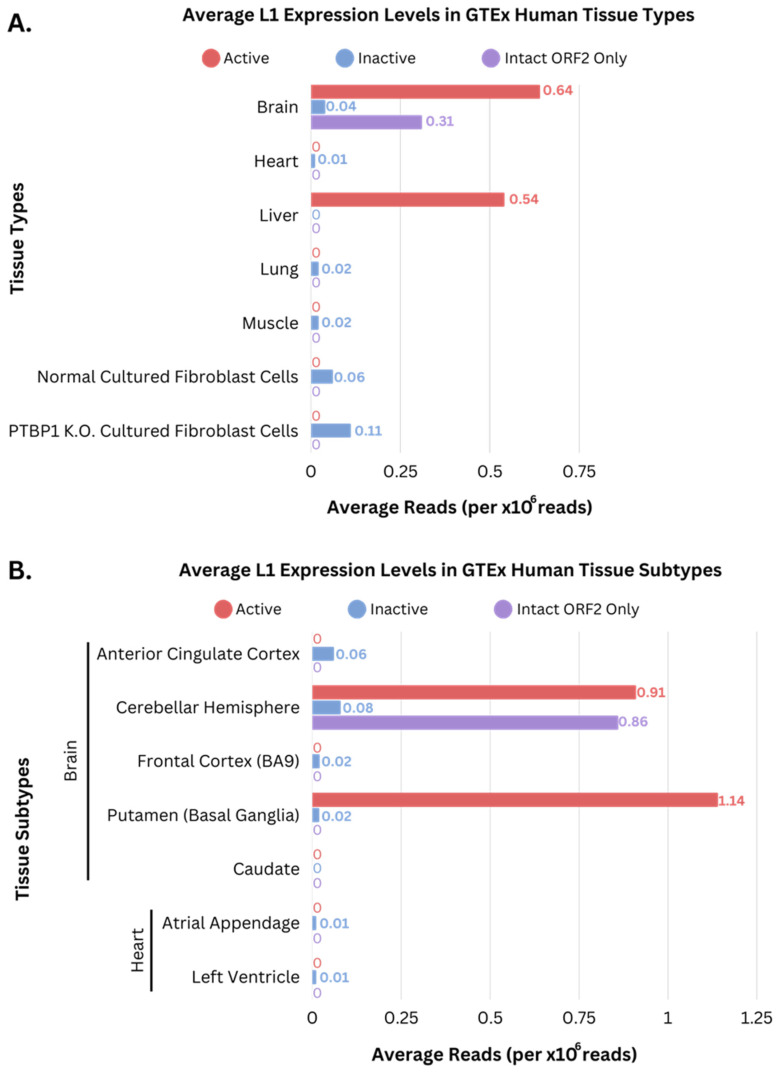
Average expression of L1 elements in 88 long-read sequenced GTEx human tissues. Average L1 expression across different human tissue samples within the active (red), inactive (blue), and intact only in ORF2 (purple) L1 categories. The general expression level of these L1 categories was determined by the mean number of reads that intersected with the L1 loci, normalized by the sample’s total read count, followed by a weighted average based on the length of the L1 element and its corresponding expression value, scaled in millions. (**A**) Average overall expression levels of L1 elements in the human tissue types from GTEx dataset. (**B**) Average overall expression levels of L1 elements in tissue subtypes from the GTEx dataset, including anterior cingulate cortex, cerebellar hemisphere, frontal cortex (BA9), putamen (basal ganglia), and caudate for brain, and atrial appendage and left ventricle for heart tissues.

**Figure 4 genes-14-01893-f004:**
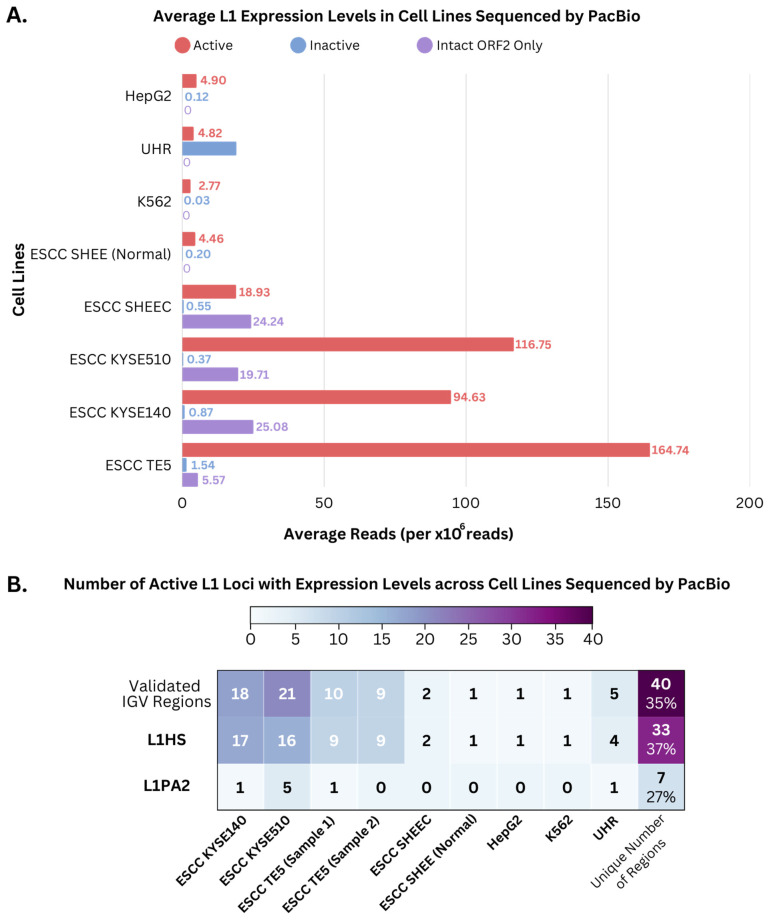
Average overall expression of L1 elements in normal and cancerous cell lines, following artifact-based filtering. (**A**) Average L1 expression for the following cell lines as sequenced by PacBio: ESCC, HepG2, UHR, and K562, within the active (red), inactive (blue), and intact only in ORF2 (purple) L1 categories. The general expression level of these L1 categories was determined by the mean number of reads that intersected with the L1 loci, normalized by the sample’s total read count, followed by a weighted average based on the length of the L1 element and its corresponding expression value, scaled in millions. (**B**) Heatmap of the number of L1 loci with expression within the active L1 regions for active L1 subfamilies, L1HS and L1PA2, across cell line samples sequenced by PacBio. The top row indicates the total number of regions manually validated in IGV and the last column displays the total number of unique L1 regions validated in IGV, including counts for both L1HS and L1PA2 subfamilies with expression evidence (top value) and the percentages of active loci for each family (bottom value). The color gradient ranges from 0 to 40, with white indicating no region and purple indicating the highest number of L1 regions.

**Figure 5 genes-14-01893-f005:**
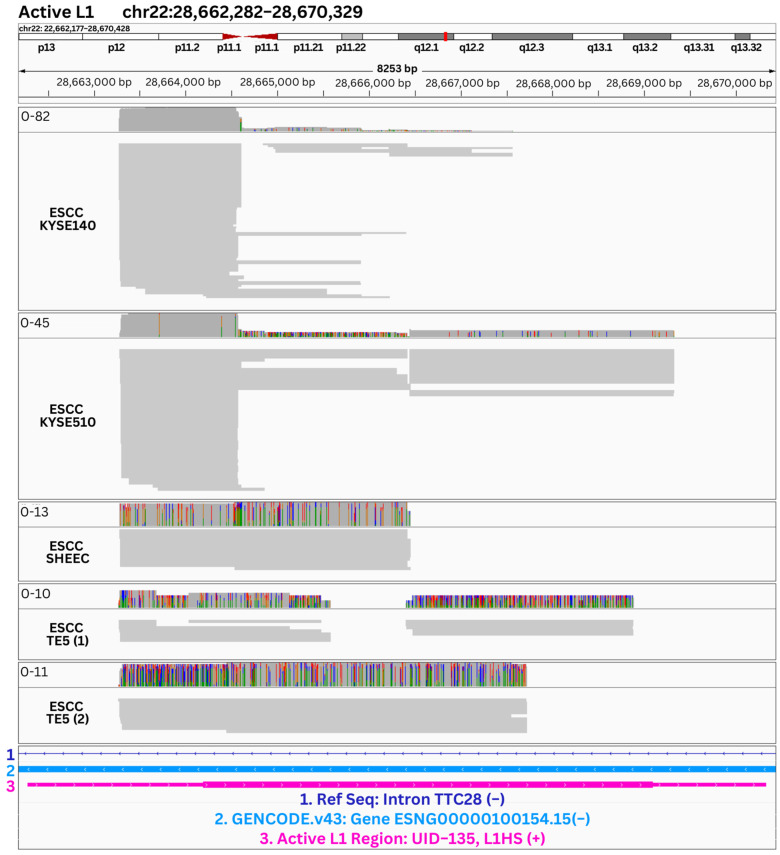
Expression of an active L1 region among cancerous ESCC cell line samples. The following samples are displayed from the top to the bottom panel: ESCC KYSE140, KYSE510, SHEEC, TE5 (sample 1), and TE5 (sample 2) at the shared active L1 region (chr22:28,662,282-28,670,329, hg38). The bottom panel shows reference annotations including, RefSeq (hg38, dark blue), GENCODE.v43 [45] (light blue) with the respective strandness, and the active L1 regions (pink) from the L1Base2 reference with the respective unique identifier (UID), subfamily name, and strandness,. For each sample, the sequencing reads are represented by grey lines, with the coverage shown in the top section.

## Data Availability

Publicly available datasets were analyzed in this study; the sources of the datasets are described in the Materials and Methods section. A reproducible workflow is available at https://github.com/WGLab/LINE-Expression-LRS.

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
