# Peer review of "Assessing the Expression of Long INterspersed Elements (LINEs) via Long-Read Sequencing in Diverse Human Tissues and Cell Lines"

_genes, 2023, doi:10.3390/genes14101893_

Round 1

Reviewer 1 Report

Dear authors,

Thanks for submitting your manuscript. In the submitted wok, you proposed and implemented a bioinformatics pipeline for the analysis of LINE expression in different human tissue and cell samples by leveraging third-generation sequencing technology. Additionally, your study compared different L1 expression patterns and provided insight into the biological functionality of L1 in different conditions.  The paper is well-written, but there are some areas that could be improved:

  1. Please add line numbers for your manuscript for easy reference in the future.
  2. Figure 2: 
    1. Data Pre-Processing: "< 1kb" should be "> 1kb"
    2. Artifact-Based Filtering: "< 90% Reads Map only to L1" does not match with the legend and is confusing
  3. The number of reads that survived the RepeatMasker step should be included in table S2.
  4. Figure 5: coverage information is not visible
  5. Inconsistency: 'only a small number of L1 elements (50-100) maintain the coding capacity' vs '146 regions, referred to as "Active" in Figure 2'
  6. The sentence "Subfamily names of L1 elements longer than 4.5kb in length from the UCSC Genome Browser (hg38) RepeatMasker Track were used to determine the L1 subfamily breakdown in the active, full-length L1 loci." needs to be clarified.
  7. Since you have built a pipeline for this type of analysis, it can be beneficial to include a few neurological disease samples in the current study to further investigate the relationship between LINE and neurological disease.
  8. Regarding the pipeline:
    1. It is highly recommended to adopt a workflow management engine such as Nextflow or Snakemake to chain your scripts, which will make your pipeline easier to deploy and scale up.
    2. In your GitHub repo, it would be helpful to include an example dataset, example commands and main output plots/tables.
    3. Will a table summarizing the read counts after each steps be created by the pipeline?

Thank you for your attention to these suggestions.

Author Response

Thank you very much for taking the time to review this manuscript, we greatly appreciate your comments towards its improvement for publication. Please find the detailed responses below and the corresponding revisions/corrections highlighted in the re-submitted files.

Comments 1: Please add line numbers for your manuscript for easy reference in the future.

Response 1: Thank you for the comments. Line numbers have been included in the most recent re-submitted draft.

Comments 2: Figure 2: Data Pre-Processing: "< 1kb" should be "> 1kb".

Response 2: Thank you for your correction, the change from '< 1kb' to '> 1kb' has been implemented into Figure 2.

Comments 3: Artifact-Based Filtering: "< 90% Reads Map only to L1" does not match with the legend and is confusing.

Response 3: Thank you for pointing this out, we have changed "< 90% Reads Map only to L1" to “Reads with < 90% mapping to the L1 loci”. Additional clarification has also been included within the text in lines 222-227 using track changes. 

Comments 4: The number of reads that survived the RepeatMasker step should be included in table S2.

Response 4: Thank you for your comment, we agree that including the number of reads that survived the RepeatMasker step in table S2 would be beneficial for clarity about the pipeline method. This information is now integrated.

Comments 5: Figure 5: coverage information is not visible.

Response 5: Thank you for your feedback, we have updated the coverage information on Figure 5 to be more visible.

Comments 6: Inconsistency: 'only a small number of L1 elements (50-100) maintain the coding capacity' vs '146 regions, referred to as "Active" in Figure 2'.

Response 6: Thank you for pointing out the inconsistency, we have revised the statement to maintain clarity and accuracy. The revised sentence now reads: 'only a small number of L1 elements maintain the coding capacity' vs '146 regions, referred to as "Active" in Figure 2'.

Comments 7: The sentence "Subfamily names of L1 elements longer than 4.5kb in length from the UCSC Genome Browser (hg38) RepeatMasker Track were used to determine the L1 subfamily breakdown in the active, full-length L1 loci." needs to be clarified.

Response 7: Thank you for your feedback, we've clarified the sentence by adding the specific subfamily names, “L1HS and L1PA2” to line 226.

Comments 8: Since you have built a pipeline for this type of analysis, it can be beneficial to include a few neurological disease samples in the current study to further investigate the relationship between LINE and neurological disease.

Response 8: Thank you for your insightful suggestion on analyzing additional neurological disease samples using our pipeline. Evaluating the influence of LINE elements in brain tissue in neurological disorders is indeed one of our goals for future research as we continuously strive to improve our pipeline and make it more accessible to the broader research community. To this end, we have incorporated future research directions within the Conclusion section as we hope the readers will be motivated to apply our pipeline to answer intriguing questions related to neurological diseases. In this manuscript, we focused on the development of a novel bioinformatics pipeline on profiling and quantifying full-length LINEs and proved its utility in long-read RNA-seq of diverse human tissues and cell lines. We believe that the novel computational pipeline and quantification results presented in this manuscript stand on their own to demonstrate a significant advance in the field of TE. To reflect these future research directions mentioned, we have added the following to the conclusion, “As one of our future research directions, we plan to examine L1 elements in putamen samples from Parkinson’s disease patients and compare with L1 expression in normal putamen samples. This is motivated by our observation of substantial coverage of active L1 regions in the putamen, which is implicated in Parkinson’s disease pathology.”.

Regarding the Pipeline:

Comments 9: It is highly recommended to adopt a workflow management engine such as Nextflow or Snakemake to chain your scripts, which will make your pipeline easier to deploy and scale up.

Response 9: Thank you for your suggestion to enhance the performance of our pipeline, your suggestion would help in streamlining and scaling up the computational workflow. While we may not be able to address this recommendation for the current submission due to the short timeline, we will certainly consider it for future iterations and improvements of our pipeline.

Comments 10: In your GitHub repo, it would be helpful to include an example dataset, example commands and main output plots/tables.

Response 10: Thank you for your suggestion, we agree that including an example dataset would be beneficial and have included the PacBio sequenced publicly available Universal Human Reference (UHR) dataset as an example (Please note that this particular dataset is publicly available for direct downloading). Each of the scripts have the exact commends necessary to carry out each of the steps, including example execution commands, which depends on the HPC the user has, as well as additional details regarding the expected output files.

Comments 11: Will a table summarizing the read counts after each step be created by the pipeline?

Response 11: Thank you for your question, the corresponding read count outputs are already incorporated into the pipeline for each sample following LongReadSum quality check analysis. We provide the LongReadSum output, which contains the read counts after each step, stored within the LongReadSum folder in the summary file. We provide the output following mapping to the hg38 reference genome and artifact-based filtering. 

Reviewer 2 Report

In this research paper, authors developed a novel bioinformatic pipeline for the quantification of LINE elements in human genome. To address this issue, authors take advantage of plethora of various long-read RNA-sequencing datasets from human tissues and cell lines. A software developed allowed authors to estimate the expression of diverse L1 transposable elements in a genome-wide scale. Source code of software is also published with well-documented step-by-step manual in github. Authors also applied a manual curation protocol for expressed LINE loci, revealing different read alignment patterns. To summarize, authors are presenting full-fledged and engaging research that will be of interest to the scientific community, however, i've got a few comments.

1) In the sentence "Using the RepeatMasker output, sequences containing L1 elements that were identified as less than 10% diverged from the consensus sequence were selected for the downstream analysis", specifically, what variable of RepeatMasker did you use? "div" column? 
2) What is the level of diversity (for example, standard deviation) among various biological samples concatenated?  

Author Response

Thank you very much for taking the time to review this manuscript, we greatly appreciate your comments towards its improvement for publication. Please find the detailed responses below and the corresponding revisions/corrections highlighted in the re-submitted files.

Comments 1: In the sentence "Using the RepeatMasker output, sequences containing L1 elements that were identified as less than 10% diverged from the consensus sequence were selected for the downstream analysis", specifically, what variable of RepeatMasker did you use? "div" column?

Response 1: Yes, the “div” column from the RepeatMasker output was used as the filtering criteria, retaining elements that were less than 10% diverged. This information was added to the main text in line 171.

Comments 2: What is the level of diversity (for example, standard deviation) among various biological samples concatenated? 

Response 2: In the case of our GTEx dataset, the samples are derived from different individuals, not necessarily multiple tissues from the same person. Therefore, assessing the level of diversity, such as standard deviation, among the tissues does not apply. These samples are distinct biological samples, rather than biological replicates, and therefore this assessment does not apply to the dataset used.

Reviewer 3 Report

In the manuscript by Rybacki et al., the authors developed a bioinformatics pipeline using advanced sequencing technologies to identify and quantify active, full-length LINEs, focusing on young L1 elements with intact retrotransposition machinery, and found that active LINEs are expressed in cancer, particularly at intact L1 loci. The manuscript is well-written and the data well organized. One minor comment is that in Figure 5 legend, the authors should give an explanation for the substantial mismatches in the IGV coverage tracks. Are those due to low accuracy of long-read sequencing data or other reasons?

Author Response

Thank you very much for taking the time to review this manuscript, we greatly appreciate your comments towards improving the manuscript for publication. Please find the detailed responses below and the corresponding revisions/corrections highlighted in the re-submitted files.

Comments 1: One minor comment is that in Figure 5 legend, the authors should give an explanation for the substantial mismatches in the IGV coverage tracks. Are those due to low accuracy of long-read sequencing data or other reasons?

Response 1: Thank you for your comment, these mismatches are primarily attributed to the nature of the PacBio Iso-Seq data. We see that the first two samples in Figure 5 have less mismatches as they contain shorter fragments. It is important to note that shorter reads tend to have higher accuracy, as they benefit from consensus-based error correction during IsoSeq analysis, and therefore exhibit a lower number of mismatched bases in the IGV plot for these samples. Additionally, it's worth considering the possibility of a polymorphic L1 element that may not be present in the reference genome, which could also contribute to the observed variation in mismatch patterns in Figure 5. We have also clarified this within the main text of the manuscript in lines 429-436.

Author Response

Thank you very much for taking the time to review this manuscript, we greatly appreciate your comments towards improving the manuscript for publication. Please find the detailed responses below and the corresponding revisions/corrections highlighted in the re-submitted files.

Comments 1: In line 12 of page 8, the authors mention filtering out reads that were less than 1kb long. Could you please provide the scientific basis or rationale for this date filtration?

Response 1: Thank you for your comment, the rationale for this filtration method was grounded in the structure and functional characteristics of LINE elements. These elements are known for their longer lengths, around 6-8kb, and those less than 1kb in length are considerably shorter than the usual length of LINE elements, suggesting a lack of the necessary components for retrotransposition. We also removed these short sequences to help performance of the pipeline, as well as noise to help focus the analysis on full-length LINEs. In the revised manuscript we further clarified the rationale by clarifying the sentence in line 154-156.

Comments 2: In line 16 of page 8, the authors mention’ Throughout our analysis, we anticipated that these variations would inherently result in variable L1 expression levels, since these technical variations can have an influence on recapitulation of L1 transcripts that are longer than typical mRNA transcripts’. It is imperative that the authors furnish more comprehensive information regarding these technical variations, elucidating their precise characteristics and their direct implications on the observed variability in L1 expression levels. Such clarification is vital for rigorous evaluation of the experimental results and their implications.

Response 2: Thank you for highlighting the concern regarding the technical variations and their potential impact on L1 expression levels. It is important to note that our pipeline is designed to be generalized as it can accommodate various datasets with different technical characteristics. We first further address the more recent changes in Nanopore’s error profile in line 368-369, “With the recent introduction of R10.4.1 flowcell with the sequencing kit 14, the sequencing error rate has dropped to 1%.”. We then describe the technical variations later in the discussion, specifically under section “3.4 Diverse Sequencing Statistics and their Differential Impact on L1 Profiling”, starting from line 511 we explain when we say “technical variation” we are referring to the sequencing and library construction methods, and how these aspects can vary and subsequently affect the results. We hope these explanations offer a more comprehensive understanding of how technical variations were considered within the scope of our analysis.

Comments 3: Figure 4B compares the number of active L1 loci in different cell lines for L1HS and L1PA2. However, the total counts in these groups appear to be dissimilar. To provide a more intuitive representation of the experimental results, it is recommended to normalize these counts. Please consider normalizing the data to account for the varying total counts, which will enhance the clarify and fairness of the comparisons in the figure.

Response 3: Thank you for your suggestion, we have revised Figure 4B to improve the clarity in the representation of the results and account for varying total counts by normalization. In addition to the number of loci in each subfamily, we now also included the percentage of active loci that showed evidence of expression in Figure 4B. The manuscript has been revised in the caption for Figure 4B, lines 403-404.